# A Study on the Effectiveness of VR Rehabilitation Training Content for Older Individuals with Total Knee Replacement: Pilot Study

**DOI:** 10.3390/healthcare12151500

**Published:** 2024-07-29

**Authors:** DooChul Shin, SoungKyun Hong

**Affiliations:** 1Department of Physical Therapy, College of Health and Welfare, Sahmyook University, Seoul 01795, Republic of Korea; icandox@syu.ac.kr; 2Department of Physical Therapy, College of Health Science, Woosuk University, Wanju 55338, Republic of Korea

**Keywords:** VR, knee joint, osteoarthritis, arthroplasty, post-exercise recovery

## Abstract

There is a paucity of research applying fully immersive virtual reality (VR) training to older adults with degenerative joint disease. This study investigated the effects of a training program utilizing fully immersive VR games on proprioception and gait ability in older patients with degenerative arthritis who had undergone total knee arthroplasty. This randomized controlled trial enrolled patients aged ≥65 years who were diagnosed with knee joint arthritis and had undergone knee arthroplasty followed by physical therapy. Participants were randomly assigned to an experimental group (receiving training using fully immersive VR games along with regular physical therapy) and a control group (receiving only regular physical therapy). The intervention was conducted five times a week for four weeks. Knee joint proprioception was measured using Biodex before and after the intervention. Spatial–temporal gait variables were collected using OptoGait for gait assessment. There was a significant decrease in the absolute error values of proprioception after the intervention in the experimental group, compared to before (*p* < 0.05), indicating improvement in proprioception. Gait speed, step count, and stride length improved significantly (*p* < 0.05, *p* < 0.01), demonstrating an enhancement in gait ability. The experimental group showed significantly greater improvements in gait speed, step count, and stride length than the control group (*p* < 0.01). Training using a fully immersive VR exercise program may have potential benefits for improving proprioception and gait parameters in patients who have undergone total knee arthroplasty. Fully immersive VR game-based training can be utilized as an effective rehabilitation intervention for patients undergoing knee arthroplasty in the future.

## 1. Introduction

As living standards and medical technology have improved worldwide, life expectancy has increased. Many older adults suffer from various chronic diseases, among which osteoarthritis (OA) is common, affecting up to 53% of the population aged 65 and above and significantly decreasing their quality of life [1]. The incidence of degenerative joint disease is continually rising globally, impacting older adults’ health due to pain, joint function impairment, and decreased quality of life [2]. The knee joint, which bears more weight than other joints, shows a higher prevalence of degenerative joint disease, at 82.6% compared to other joints [1].

Degenerative joint disease of the knee leads to muscle imbalance, increased pain, muscle weakness, and decreased range of motion [3]. As knee OA progresses, it decreases muscle strength and balance, impairing walking ability and increasing fall risk among older individuals [4].

Treatment options for knee OA include pharmacological, non-pharmacological, and surgical interventions. When conservative treatments fail, knee arthroplasty, a surgical intervention, is commonly used to relieve pain, improve joint mobility, and maintain alignment and stability [5]. However, postoperative complications. such as pain, reduced joint mobility, muscle weakness, decreased proprioception, balance impairment, and walking difficulties, can occur [5]. Therefore, early exercise initiation is recommended to alleviate pain, increase joint mobility, and enhance the functional performance of the knee joint for optimal surgical outcomes [6].

Common exercise programs for patients undergoing knee arthroplasty include range of motion exercises, proprioceptive neuromuscular facilitation, sling exercises, obstacle-course walking, and stability exercises. However, more systematic and quantitative comparisons based on recent research are needed for optimal intervention [7,8,9]. The development of scientific exercise rehabilitation programs incorporating new technologies. such as virtual environments, has been continuously emphasized [10].

Recently, VR rehabilitation training has been applied to patients with neurological and musculoskeletal disorders, providing intrinsic motivation and feedback through visual, auditory, and tactile sensations, thereby enhancing exercise engagement and effectiveness [11]. Studies by Lee et al. demonstrated the positive effects of VR exercise programs on balance and muscle strength in patients with Parkinson’s disease [12]. Shin et al. reported positive effects on lower limb strength and balance in brain injury patients through VR rehabilitation exercises [13]. Moreover, Hong divided patients who had undergone knee arthroplasty into control and experimental groups, applying continuous passive motion (CPM) therapy and exercise therapy to the control group and adding VR exercise programs to the experimental group, confirming the effects of VR on muscle strength, balance, and walking ability [14]. However, previous studies using VR have mainly targeted patients with central nervous system injuries and older individuals in single-patient studies. Research applying fully immersive VR training to older patients with degenerative joint disease, who constitute the majority of knee arthroplasty candidates, is particularly scarce.

Therefore, this study aimed to investigate the efficacy of a rehabilitation training pro-gram using fully immersive VR games on proprioception and gait variables in older patients with degenerative joint disease who had undergone a total knee arthroplasty.

## 2. Materials and Methods

### 2.1. Participants

This study was conducted after approval was obtained from the Institutional Review Board of Woosuk University (WS-2023-06). All participants received an explanation of the study and gave their consent before participating. From June 2023 to November 2023, the study was carried out on 16 older patients with degenerative joint disease who had undergone a total knee arthroplasty at Hospital B in Busan, South Korea. They were recruited through hospital advertisements. The inclusion criteria were individuals aged 65 or older diagnosed with moderate to severe degenerative joint disease who had undergone a unilateral knee arthroplasty, were capable of independent walking for at least ten meters without assistive devices one week after surgery, and who could understand the study’s purpose and methods and sign the informed consent form. The exclusion criteria were as follows: individuals with a history of a previous knee arthroplasty; those with postoperative infections; individuals experiencing pain in areas other than the knee; and those using medications affecting their balance or muscle strength.

### 2.2. Research Procedures

This preliminary study compared the effects of a training program using immersive VR games with general physical therapy methods in older patients with knee osteoarthritis who had undergone a total knee arthroplasty. The study aimed to elucidate the effects of an immersive VR game-based training program on knee joint proprioception and gait. The experimental design flowchart is presented in Figure 1. Participants were randomly assigned to an experimental group using fully immersive VR and a control group using only general physical therapy. Both the experimental and control groups were tested for knee-joint reproduction angle proprioception and for temporal and spatial gait variables both before and after the training to confirm any changes. Pre-tests and post-tests were conducted one day before the start of the intervention and the day after the intervention ended. All pre- and post-tests were administered by the same therapist, who was unaware of the study participants’ group assignments. All participants were instructed to engage in continuous passive motion (CPM) therapy for 30 min daily, starting from the first day after surgery. CPM was performed on all participants for only one week. Additionally, after the first week post-surgery, the experimental group received 20 min of training using fully immersive VR games immediately after 30 min of general physical therapy [14]. The virtual training sessions were divided into two 10 min sessions, with a 5 min rest interval in between to prevent fatigue and dizziness. The control group received only general physical therapy for 50 min. All interventions were conducted five times a week for four weeks.

### 2.3. Proprioception

For knee joint proprioception measurement, the absolute error was measured using the Biodex medical system (Ⅵ Pro, Biodex Medical Systems, Inc., Shirley, NY, USA) (Figure 2). Participants’ vision and hearing were blocked, and the knee angle was restricted within a range of full extension (0° to 90° flexion), with reproduction of the perceived angle of 30° flexion from 90° flexion.

The starting position was always matched to 90° flexion. Participants were instructed to perceive the target angle for five seconds, return to 90° flexion upon receiving a readiness signal, and then attempt to return to the target angle. Measurements were repeated three times, and the error values between the target angle and the measured angle were calculated. The test–retest reliability of this assessment was ICC = 0.85 [15].

### 2.4. Gait Ability

To analyze gait, a gait analyzer (OPTOGait, Microgate S.r.I, Bolzano, Italy) was employed. (Figure 3). Using the gait analysis system, participants’ gait speed, cadence, and stride length were recorded through computerized analysis. Participants were instructed to stand at the starting point of the gait analysis system and walk at a comfortable pace for five meters, synchronized with the signal from the examiner’s shoe. The test–retest reliability of this tool showed high reliability, with ICC = 0.93 to 0.99 [16].

### 2.5. Training Using a Fully Immersive VR Game

The fully immersive VR gaming equipment consisted of the Meta Quest 2 (CUH-7117B, Meta, Menlo Park, CA, USA). This equipment includes a head-mounted display (CUH-ZVR2, Meta, Menlo Park, CA, USA) and motion controllers (CECH-ZCM2G, Meta, Menlo Park, CA, USA). Additionally, a 72-inch LCD screen was used as the visual display device. The visual display device allowed the therapist applying the intervention to view the screen that the study participant was viewing using the head mount. The participants performed tasks in a VR environment that changed in real time, based on the movements of their heads after wearing the head-mounted display in a spacious area free from the risk of falls. Through the head-mounted display, participants observed the virtual environment that responded in real time to their head movements and the movements of the motion controllers held in their hands. Additionally, the auditory effects provided through the headphones attached to the head-mounted display further enhanced the sense of realism experienced by the participants. The game utilized in the training was Fruit Ninja, developed by Sony Japan. In Fruit Ninja, as participants wearing the head-mounted display move their heads, motion sensors detect these movements, causing the in-game perspective to shift accordingly. Additionally, the sensors recognize the movements of the controllers held in the participants’ hands. In the VR environment, participants were tasked with slicing fruit approaching from the left and right sides using controllers, which were recognized as swords. This activity enabled participants to engage in lower body exercises involving weightbearing and weight shifting through movements of the lower extremities (Figure 4).

### 2.6. Conventional Physical Therapy

As with conventional physical therapy, passive knee joint mobilization by the therapist and active range of motion exercises by the participants were performed for 10 min. Q-setting exercises and isometric exercises of the quadriceps were carried out for 10 min; balance training on a balance pad was conducted for five minutes, and gait training using parallel bars was conducted for five minutes.

### 2.7. Statistical Analysis

Data were analyzed using SPSS 22.0 software (IBM, Chicago, IL, USA). All data were analyzed for normality using the Shapiro–Wilk test. A paired *t*-test was performed to compare the independent variable before and after training within the group. An independent *t*-test was performed to compare the differences in the dependent variable, according to the intervention between groups. A *p*-value <0.05 was statistically significant.

## 3. Results

This study randomly assigned patients undergoing total knee arthroplasty to an experimental group and a control group, applying a 4-week intervention. The general characteristics of the participants are provided in Table 1.

The experimental group showed a statistically significant decrease in the absolute error of proprioception from 4.58° before intervention to 2.91° after intervention (*p* < 0.05). The control group showed a decrease in the absolute error of proprioception from 6.73° before the intervention to 4.83° after the intervention; however, this difference was not statistically significant, and no significant differences based on the intervention method were observed between the groups (Table 2).

The experimental group exhibited a statistically significant increase in gait velocity from 0.87 m/s before the intervention to 0.92 m/s after the intervention (*p* < 0.05). In contrast, the control group showed a statistically significant decrease in gait velocity from 0.96 m/s before the intervention to 0.89 m/s after the intervention (*p* < 0.01). A significant improvement was observed in the experimental group compared to the control group, based on the intervention method (*p* < 0.01).

The experimental group showed a statistically significant increase in cadence, from 99.27 steps/min before the intervention to 103.99 steps/min after the intervention (*p* < 0.01). The control group exhibited a decrease from 102.45 steps/min before the intervention to 99.00 steps/min after the intervention; however, this difference was not statistically significant. Additionally, a significant improvement was observed in the experimental group compared to the control group, based on the intervention method (*p* < 0.01).

The experimental group demonstrated a statistically significant increase in stride length, from 47.44 cm before the intervention to 50.19 cm after the intervention (*p* < 0.01). In contrast, the control group showed a decrease from 50.30 cm before the intervention to 48.55 cm after the intervention; this difference was statistically significant (*p* < 0.05). Furthermore, a significant improvement was observed in the experimental group compared to the control group, based on the intervention method (*p* < 0.01) (Table 3).

## 4. Discussion

This study investigated the effectiveness of fully immersive VR rehabilitation program on the proprioception of knee joint angle reproduction and gait variables. The participants were older individuals with degenerative knee joints who had undergone total knee arthroplasty. The results showed a positive difference in proprioception and gait variables before and after the intervention in the experimental group that received the fully immersive VR exercise program.

Proprioception, the ability to reproduce the desired angle of a joint through sensory input, plays a crucial role in motor functioning, including balance and vestibular senses. It provides information to the associated areas to induce and facilitate both voluntary and involuntary movements [17]. However, patients with knee osteoarthritis have a reduced proprioceptive sense of joint position, and there is an even greater reduction in joint position sense after knee arthroplasty [18]. This deficiency in proprioceptive sense can lead to joint instability, increasing the risk of injuries from, for example, fall down. [19]. Combining general physical therapy with fully immersive VR gaming in this study showed the potential for enhancing proprioceptive sense in patients undergoing total knee replacement. These results suggest that the immediate visual feedback provided may facilitate joint realignment, thereby activating the proprioceptive sense responsible for joint position perception. The immediate feedback in this study referred to recognizing flying fruit with the eyes, rather than feedback on lower extremity movements. The movement of lower extremities may have been promoted through weight shifting or trunk rotation in response to visual feedback.

This observation is supported by research, such as the study by Deblock-Bellamy et al. that reported an improvement in upper limb proprioception through VR training [20], and by previous research on peripheral nerve patients that demonstrated the positive effects of VR training on proprioceptive sense [21]. These findings lend support to the hypothesis of this study.

It is known that older adults who have undergone knee replacement surgery experience a decrease in walking speed by approximately 69% compared to their healthy counterparts [22]. This is due to pain, muscle weakness, and decreased proprioception, resulting in a slower walking speed. These symptoms restrict knee flexion in both the stance and swing phases [23], ultimately leading to a shorter stride length [24]. The decline in proprioception diminishes the ability to place the foot accurately, causing the body to adopt cautious walking patterns for stability and fall prevention. [25]. Consequently, not only does walking speed decrease, but stride length and cadence also decrease.

The study suggests that the combined use of general physical therapy and fully immersive VR gaming training may positively contribute to improving gait parameters in patients who have undergone knee arthroplasty. These results likely stem from improved proprioceptive sensitivity, leading to enhanced muscle coordination and balance abilities. Consequently, the restoration of shortened stance phases during walking increases stride length, resulting in improved gait speed and cadence.

These findings are supported by the results of Baram and Miller’s study, in which it was observed that VR training for patients with multiple sclerosis improved walking speed and reduced restrictions in lower limb movements [26]. Additionally, Lamontagne et al. and Yang et al. observed significant improvements in walking speed after applying VR exercise programs to stroke patients, further substantiating the positive impact of VR exercise programs on walking ability, as demonstrated in this study [27,28].

However, the control group showed a significant decrease in gait speed and step length. The elderly total knee arthroplasty patients in this study may have experienced rapid muscle-strength decline after surgery. The control group only received conservative treatment focused on pain and range of motion, leading to further muscle weakening during the 4-week period after surgery, resulting in decreased step length and walking speed.

In summary, the increased walking function observed in the experimental group after the fully immersive VR exercise programs can be attributed to enhanced proprioception, which allows for the reproduction of proper knee joint flexion angles. This increase in proprioception leads to improvements in muscle coordination and balance, resulting in a sufficient stance phase during walking, thereby increasing step length. Consequently, this leads to an overall increase in walking speed and cadence. Considering previous research findings indicating that a decrease of 10 cm/s in walking speed for older adults equates to a 10% reduction in the ability to perform daily activities [29], it is expected that the increased walking speed observed after the intervention in this study will enhance walking ability and contribute positively to the functional capacity of older adults in their daily lives.

This study investigated the efficacy of fully immersive VR rehabilitation program for older adults who had undergone total knee arthroplasty due to degenerative joint disease, focusing on proprioception and gait parameters. This preliminary study suggested that a fully immersive VR exercise program may have potential benefits for improving proprioception and gait parameters in patients who have undergone total knee arthroplasty. However, there are several limitations to this study. First, the relatively short intervention period and the small sample size make generalization difficult. Second, there were limitations in selecting content with an appropriate degree of difficulty, tailored to each individual’s level of impairment or balance ability, due to constraints in the types of available content. Therefore, considering the limitations mentioned, there is a need to develop appropriate VR content tailored to musculoskeletal rehabilitation. Future research should focus on developing diverse content and systematic VR programs, considering these limitations.

## 5. Conclusions

This pilot study aimed to explore the potential benefits of a VR rehabilitation training program for older patients with degenerative joint disease who had undergone total knee arthroplasty, particularly in terms of improving proprioception and gait variables. The results indicated that there were improvements in proprioception and gait variables among these patients. It is judged that training with fully immersive VR rehabilitation program may be effectively utilized as an intervention method for early rehabilitation in total knee arthroplasty patients.

## Figures and Tables

**Figure 1 healthcare-12-01500-f001:**
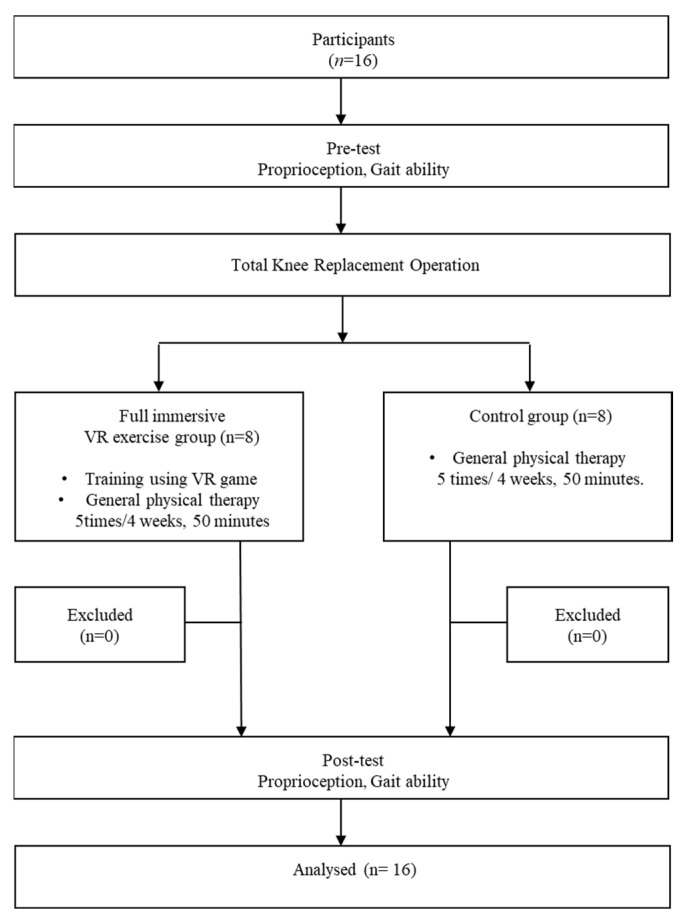
Experimental flow chart.

**Figure 2 healthcare-12-01500-f002:**
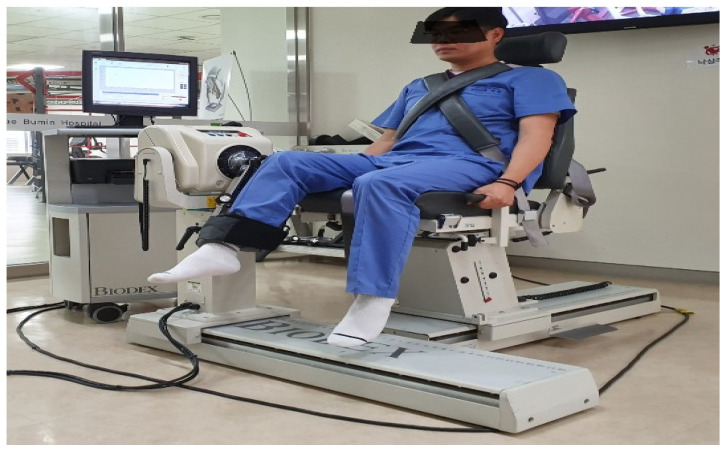
Knee joint angle reproduction test.

**Figure 3 healthcare-12-01500-f003:**
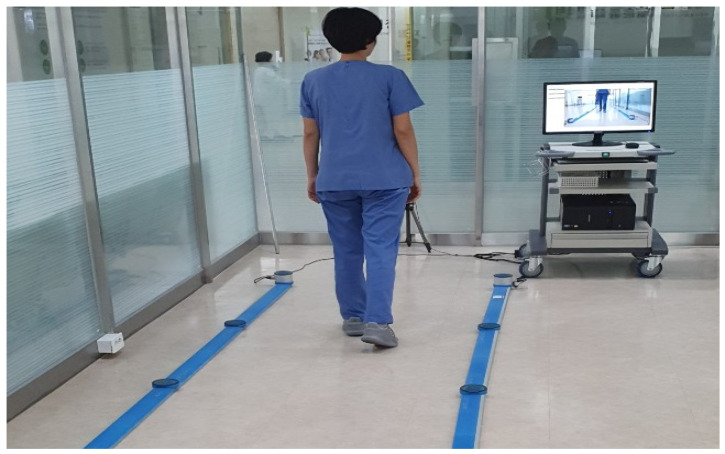
Temporal and spatial gait analysis.

**Figure 4 healthcare-12-01500-f004:**
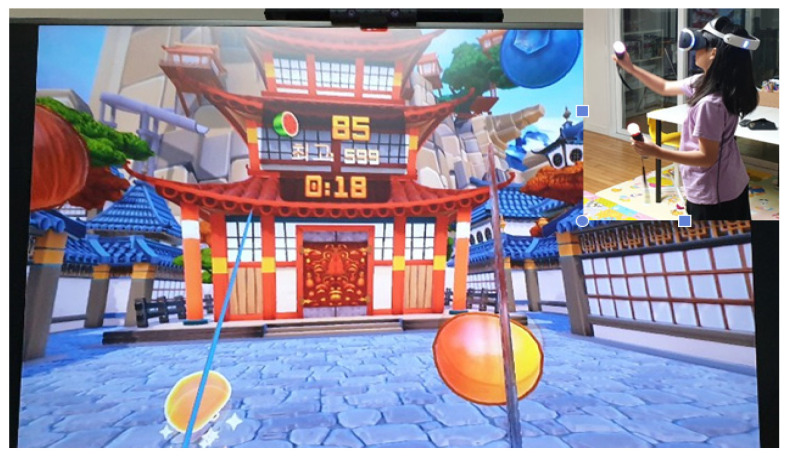
Fully immersive virtual reality game.

**Table 1 healthcare-12-01500-t001:** General characteristics of the participants (n = 16).

Characteristics	Study Group (n = 8)	Control Group (n = 8)
Age (years)	66.00 ± 2.35	67.20 ± 2.39
Height (cm)	156.60 ± 8.02	152.80 ± 2.95
Weight (kg)	56.80 ± 5.54	65.80 ± 6.72
BMI (kg/m^2^)	23.12 ± 0.75	28.29 ± 2.76

Values are presented in terms of the mean ± SD. BMI, body mass index.

**Table 2 healthcare-12-01500-t002:** Changes in proprioception.

Variables		Experimental Group (n = 8)	Control Group (n = 8)	t
Proprioception (°)	Pre	4.58 ± 2.75	6.73 ± 2.86	−1.530
	Post	2.91 ± 1.51	4.83 ± 3.44	
	Change	1.66 ± 1.84	1.90 ± 2.95	−1.930
	t	2.555 *	1.819	

Values are presented in terms of the mean ± SD (* *p* < 0.05).

**Table 3 healthcare-12-01500-t003:** Change in gait variables.

Variables		Experimental Group (n = 8)	Control Group (n = 8)	t
Gait velocity (m/s)	Pre	0.87 ± 0.13	0.96 ± 0.15	−1.351
	Post	0.92 ± 0.13	0.89 ± 0.17	
	Change	−0.05 ± 0.05	0.08 ± 0.04	−5.305 **
	t	−2.754 *	5.123 **	
Cadence (step/min)	Pre	98.27 ± 11.03	102.45 ± 16.67	−0.591
	Post	103.39 ± 8.38	99.00 ± 18.27	
	Change	−5.11 ± 3.54	3.45 ± 4.67	−4.134 **
	t	−4.087 **	2.090	
Step length (cm)	Pre	47.44 ± 5.65	50.30 ± 3.39	−1.229
	Post	50.19 ± 3.63	48.55 ± 3.20	
	Change	−2.75 ± 2.75	1.75 ± 1.50	−4.067 **
	t	−2.829 **	3.309 *	

Values are presented in terms of the mean ± SD (* *p* < 0.05, ** *p* < 0.01).

## Data Availability

The data presented in this study are available on request from the corresponding author.

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
