# Peer review of "A Study on the Effectiveness of VR Rehabilitation Training Content for Older Individuals with Total Knee Replacement: Pilot Study"

_healthcare, 2024, doi:10.3390/healthcare12151500_

Round 1

Reviewer 1 Report

Comments and Suggestions for Authors

Author Response

Reviewer 1

Article: A Study on the Effectiveness of Home-Based Virtual Reality Rehabilitation Training Content for Older Individuals with Total Knee Replacement: pilot study

Journal: Healthcare (ISSN 2227-9032)

This work presents a proposal based on immersive virtual reality (IVR) to improve the rehabilitation of gait quality after total knee prosthesis replacement in elderly people. I want to thank the authors for their work and for allowing me to read and review this manuscript. I hope my comments help make the article suitable for publication. Please find below some suggestions for improvement:

  1. Introduction

- Please review the first sentence (lines 32, 33). How chronic conditions are related to medical technology improvement?

Response: Advances in medical technology have also led to longer life expectancy and increased risk of chronic diseases (age-related) in many people's later years. I'll explain in more detail.

“As living standards and medical technology have improved worldwide, life expectancy has increased. As life expectancy has increased, many people suffer from various chronic diseases in their old age.”

- The use of acronyms is recommended if it is a term that will be repeated throughout the manuscript, like IVR.

Response: I have edited it to use abbreviations as you requested.

- The term Metaverse is described as a technology, but it is not accurate. Line 59

Response: I have modified it as per your request.

“Metaverse”  “virtual environment”

- It is recommended to review the section where IVR technology is introduced. Line 73 - 74 no reference. What is Virtual Reality? What do you mean by fully immersive?

Response: I don't think there is a need for references because there is a very small amount of research on applying immersive virtual reality technology to total knee arthroplasty. When searching for immersive virtual reality and total knee arthroplasty on Pubmed, a total of 11 papers are retrieved. Based on these results, I think we can say that there is a lack of related research. However, it is a bit difficult to explain this content with references. I hope you understand.

Immersive Virtual Reality (IVR) refers to technology that allows users to fully immerse themselves in a computer-generated three-dimensional environment.

- What do you mean by “various patient groups” (line 62), it is ambiguous?

Response: As you requested, I have described it a little more clearly.

“Various patient groups”  “patients with neurological and musculoskeletal disorders”

- Please provide a reference, line 63.

Response: Added reference as per your request

- It is recommended to expand the reference to related work, with some work published in the last 5 years on the efficacy of IVR in improving physical mobility rehabilitation.

Examples:

Fregna, G., Schincaglia, N., Baroni, A., Straudi, S., & Casile, A. (2022). A novel immersive virtual reality environment for the motor rehabilitation of stroke patients: A feasibility study. Froiers in Robotics and AI, 9. https://doi.org/10.3389/frobt.2022.906424

Gsangaya, M. R., Htwe, O., Selvi Naicker, A., Md Yusoff, B. A. H., Mohammad, N., Soh, E. Z. F., & Silvaraju, M. (2023). Comparison between the effect of immersive virtual reality training versus conventional rehabilitation on limb loading and functional outcomes in patients after anterior cruciate ligament reconstruction: A prospective randomized controlled trial. Asia-Pacific Journal of Sports Medicine, Arthroscopy, Rehabilitation and Technology, 34, 28–37. https://doi.org/10.1016/j.asmart.2023.09.002

- The home-based condition is very interesting, however, no mention is made about the rationale of it (benefit, previous studies). See:

Salisbury, J. P., Aronson, T. M., & Simon, T. J. (2020). At-home self-administration of an immersive virtual reality therapeutic game for post-stroke upper limb rehabilitation. CHI PLAY 2020 - Extended Abstracts of the 2020 Annual Symposium

Phelan, I., Carrion-Plaza, A., Furness, P. J., & Dimitri, P. (2023). Home-based immersive virtual reality physical rehabilitation in paediatric patients for upper limb motor impairment: a feasibility study. Virtual Reality, 27(4), 3505–3520. https://doi.org/10.1007/s10055-023-00747-6 on Computer-Human Interaction in Play, 114–121. https://doi.org/10.1145/3383668.3419935

Response: Added reference as per your request

  1. Method

- Including the ethical statement in the text (Method) is recommended.

Response: As per your request, we have updated the section on research ethics by adding information about the IRB and consent prior to participating in the study.

“This study was conducted after approval was obtained from the Institutional Review Board of Woosuk University (WS-2023-06). All participants in this study received an explanation of the study and gave their consent to participate before participating in the study.”

- Line 82, “approval” twice.

Response: I have modified it as per your request.

- Line 95, “prO-“

Response: I have modified it as per your request.

- Please check the commas and full stops, some seem to be wrong, lines 95, 98, 100- 101, 105 “immersive virtual reality

Response: I have modified it as per your request.

- Line 90 “sur-gery”

Response: I have modified it as per your request.

- Figure 1 caption is not under the figure. In Figure 1 there is a spelling mistake “5 times”

Response: I have modified it as per your request.

- The Research procedure is not clear. It is recommended to include more details in this section, and a relevant literature reference to support the understanding of the measure and analysis of this study.

Response: As per your request, I have described the research procedure in a little more detail and added references.

“All participants (ppts) were instructed to engage in continuous passive motion (CPM) therapy for 30 minutes daily, starting from the first day after surgery. Additionally, from one-week post-surgery, the experimental group received 20 minutes of training using fully immersive, virtual reality games immediately after 30 minutes of conventional physical therapy. The virtual training sessions were divided into two 10-minute sessions with a 5-minute rest interval in between to prevent fatigue and dizziness. The control group received only conventional physical therapy for 50 minutes, excluding continuous passive motion therapy equipment”

- Why all ppts were instructed of CPM if then you indicated that the control group was excluding of CPM?

Response: CPM was administered only for 1 week prior to randomization of study participants. CMP was no longer administered after randomization of study participants.

- It sounds contradictive that in the inclusion criteria, you indicated “The inclusion criteria […] independent walking for at least ten meters without assistive devices two weeks after the surgery” when then you indicated that “All participants were instructed to engage in continuous passive motion (CPM) therapy for 30 minutes daily, starting from the first day after surgery. Additionally, from one-week post-surgery”.

Response: The selection criteria for study participants were changed to those who were able to walk 10 meters independently one week after surgery.

- Did the experimental and control group have conventional therapy? Could it be clearly explained in the experimental flowchart (Figure 1)? It is recommended to add the timing of immersive virtual reality (IVR) training and convention training (standard care (SC)).

Response: The order of virtual training therapy and general physical therapy was randomized. The order of treatment was determined according to the patient's treatment preference.

- Be consistent with the terminology to refer to conventional care (standard care, or general physical therapy).

Response: It was unified with general physical therapy.

- The figure information about General physical therapy (30 min) did not coincident with the description in the text (Line 109 “50 minutes”)

Response: I have modified it as per your request.

- The duration of the study (Figure 1 “ times for 4 weeks”) should be also explained in more detail in the text. Could the authors indicate why it was decided to carry out the study within this time frame? - When were collect the post-test measures?

Response: In the Korean medical environment, patients are hospitalized for 4 weeks after total knee replacement surgery. Therefore, a 4-week intervention period was set.

Post-measurement was conducted the day after the 4-week intervention.

Proprioception and gait measurement

- Was any clinician involved in taking the Biomex and gait measures, how was that measure collected?

Response: All pre- and post-tests were administered by the same therapist who was unaware of the study participants groups.

- It is a bit complicated to interpret the absolute error in terms of degrees of mobility. Is it closer to 90 better?

Response: Absolute error is not the degree closer to 90 degrees, but rather the difference between the target angle specified by the therapist and the angle reproduced by the patient.

- Why were the vision and hearing of the participants blocked?

Response: Because vision and hearing can affect proprioception, it may be important to block vision and hearing, especially when assessing proprioception using joint angles.

- How long did it take to measure proprioception and gait quality, was it collected before the first rehabilitation session and part of the last one, or was it collected in a separate appointment?

Response: Additional details are provided regarding the timing of pre- and post-tests.

“Pre-test and post-test were conducted one day before the start of the intervention and the next day after the end of the intervention, respectively.”

- When participants were instructed to stand and walk?

Response: The measurement of gait variables was described in detail. Gait variables were also measured one day before the start of the intervention and the day after the end of the intervention.

- Why in the inclusion criteria participants need it to be available to walk 10 meters and then they just needed to walk 5 meters?

Response: The minimum walking length of the equipment measuring gait variables was 5 meters. A length of 10 meters was required to clearly measure the walking ability of the study participants.

 - Why was relevant to use the TV visualization?

Response: TV visualizations are intended for use by therapists applying interventions to view what the research participant sees while wearing a head mount.

- Why was the Fruit Ninja game chosen and not another?

Response: This activity enabled participants to engage in lower body exercises involving weight-bearing and weight shifting through movements of the lower extremities. That's why I thought this game was suitable. I also thought it would be enough to motivate people to participate in the study through fun.

- In which way does this game motivate the patient to move the lower limb extremities?

Response: In the VR environment, participants were tasked with slicing fruit, approaching from the left and right sides using controllers, which were recognized as swords. This activity enabled participants to engage in lower body exercises involving weight-bearing and weight shifting through movements of the lower extremities

- Is there any measure or evidence that demonstrated that ppts are exercising the affected lower limb during the game?

Response: We conducted this experiment to prove it. There had been no previous experiments using this game on patients identical to the participants in this study.

Analysis

- Check the correct writing of statistical symbols, ex. p = p.

Response: I have modified it as per your request.

  1. Results

- It is not indicated whether an analysis has been carried out to check that both groups were equal at baseline. The results show that Proprioception measures are higher at baseline in the control group. Could the authors explain why this difference could be due to this? Is this difference significant?

Response: The homogeneity test of the subjects is shown in the table. The control group was higher in proprioception, but it was not statistically significant. You can refer to the t value in the table. In addition, it is presumed that the reason why the value was higher in the control group but there was no statistical difference is because the standard deviation values ​​are all large.

- Is the statement made in the lines 189 and190 refers to gait velocity?

Response: Yes, that's right. The sentence about gait velocity continues from the previous sentence.

  1. Discussion

- It is recommended to include the information in lines 214-217 regarding the proprioception in the method section or introduction to understand better that measure and the relevance for the study.

Response: As per your request, I have described proprioception in a bit more detail.

“This study investigated the effectiveness of a home-based, fully immersive VR rehabilitation program on proprioception of knee joint angle reproduction and gait variables.”

- It would be relevant to add qualitative data (ex. interview with patients and clinicians).

Response: Sorry, there is no relevant qualitative data.

- Regarding the statement made in lines 225-227, how in this study patients receive immediate feedback on the movement of their lower limb.

Response: The immediate feedback through the game in this study is not feedback on the movement of the lower extremities of the research participants. The feedback mentioned in the conclusion is that the flying fruit can be recognized immediately with the eyes. And the function of the lower extremities may have been promoted through movement through weight shift or trunk rotation in accordance with such visual feedback.

- The results of this study indicated that, in the control group, the conventional therapy does not seem to have a positive impact on the parameters of gait quality. Rather, the results appear to be worse in the post-intervention assessment. Could the authors explain why the conventional therapy group does not improve but worsens in these parameters?

Response: As per your request, we have provided additional information on the worsening of gait parameters in the control group.

“However, the control group showed a significant decrease in gait speed and step length. The elderly total knee arthroplasty patients who participated in this study may show rapid muscle strength decline after surgery. However, the control group only received conservative treatment focused on pain and ROM. As a result, it is thought that their muscle strength weakened further during the 4-week period after surgery, resulting in a decrease in step length and walking speed.”

Reviewer 2 Report

Comments and Suggestions for Authors

A Study on the Effectiveness of Home-Based Virtual Reality 2 Rehabilitation Training Content for Older Individuals with Total Knee Replacement: pilot study

Abstract:

Give a brief introduction and then starts with the project purpose.

Introduction:

Line 42: ‘’decreased range of motion in the surrounding muscles’’

Is the decrease of ROM happening in the joint or muscles? Adjust please.

Line 46-49: Please add reference for your statement.

Line 47-49: Clarify that the use of surgical intervention comes after the conservative treatments fails … etc

There is a misuse of references. Many sentences are not supported with references. Must be added.

Line 67: Who is Hong? An Author. So, please adjust.

Methods:

Line 82: The word ‘’approval’’ is repeated, remove the second one.

Have authors performed power calculation? Please explain.

Line 100-101: Starting from ‘’Both the experimental … ’’. Repetition and punctuation mistake. Please adjust.

Authors should report the protocol followed after TKR surgery, for both experimental and controls groups.

No heading for figure 1- moved somewhere.

Figure 1 is low quality

Check for typos in figure 1

Overall, recheck figure 1

Figure 2, 3 and 4: Clarify more in the heading

Line 142: ‘’TV’’ or screen? Please adjust.

The fully immersive virtual reality gaming equipment was provided by the researchers to the participants or the participants themselves brought it? Please clarify

Results:

Table 1: Where there any significant differences between these characteristics? Please add to the table.

Clarify that the duration you mean after intervention is four weeks, to remind the reader and remain on track.

Discussion:

Virtual reality home based program. It is only mentioned in the title and the discussion, but never in the methods, so please clarify that in the methods.

Comments on the Quality of English Language

Minor

Author Response

Reviewer 2

Abstract:

Give a brief introduction and then starts with the project purpose.

Response: I have made corrections as per your request.

“There is a paucity of research applying fully immersive VR training to older adults with degenerative joint disease.”

Introduction:

Line 42: ‘’decreased range of motion in the surrounding muscles’’

Is the decrease of ROM happening in the joint or muscles? Adjust please.

 Response: I have made corrections as per your request.

“decreased range of motion in the joints”

Line 46-49: Please add reference for your statement.

 Response: Here is a quote from a reference like the sentence below. I added the reference as per your request.

  1. Park, G.H.; Kim, T.W.; Song, H.B. Effect of knee stabilization exercise on balance and walking ability in patients with total knee replacement. Journal of Korean Academy of Orthopaedic Manual Therapy 2021, 27, 69–76.

Line 47-49: Clarify that the use of surgical intervention comes after the conservative treatments fails … etc

Response: I have added relevant information as per your request.

 If conservative treatments fail, Knee arthroplasty, a surgical intervention, is the most common method for treating knee OA, aiming to relieve pain, improve joint mobility, and maintain alignment and stability

-There is a misuse of references. Many sentences are not supported with references. Must be added.

 Response: As per your request, I have checked the references and added references for any missing parts.

- Line 67: Who is Hong? An Author. So, please adjust.

 Response: The author of this reference is the same as the author of this study. This study is a follow-up study of the author's previous pilot study.

Methods:

- Line 82: The word ‘’approval’’ is repeated, remove the second one.

 Response: I have made corrections as per your request.

-Have authors performed power calculation? Please explain.

 Response: Unfortunately, this study was not conducted with an appropriate sample size calculated. This is because this study was conducted after surgery in a clinical setting, and it was difficult to secure a large number of subjects. These limitations of this study are described in the limitations of the study.

“However, there are several limitations to this study. First, the relatively short intervention period and the small sample size make generalization difficult.”

-Line 100-101: Starting from ‘’Both the experimental … ’’. Repetition and punctuation mistake. Please adjust.

Response: I have made corrections as per your request.

-Authors should report the protocol followed after TKR surgery, for both experimental and controls groups.

 Response: In this study, rehabilitation after total knee arthroplasty was not performed according to a special protocol. In Korea, rehabilitation after surgery is generally performed according to the doctor's prescription. The therapist cannot decide this alone. As a result, rehabilitation after surgery was performed according to the doctor's prescription rather than according to a special protocol. In this study, rehabilitation after surgery was performed after prior consultation with the doctor.

-No heading for figure 1- moved somewhere.

Response: I have made corrections as per your request.

Figure 1 is low quality, Check for typos in figure 1Overall, recheck figure 1

Response: I have made corrections as per your request.

Figure 2, 3 and 4: Clarify more in the heading

 Response: I have made corrections as per your request.

Line 142: ‘’TV’’ or screen? Please adjust.

Response: I have made corrections as per your request.

The fully immersive virtual reality gaming equipment was provided by the researchers to the participants or the participants themselves brought it? Please clarify

Response: The VR game used in this study was provided by the researcher. Because of the nature of this study, it is not important who provided the relevant games, so the relevant content was not mentioned in the text. This part was sufficiently explained at the stage of consent of the research subjects before starting the study. In addition, most studies generally receive the relevant intervention equipment and software from the researcher. The research participants do not directly purchase these and participate in the study.

Results:

Table 1: Where there any significant differences between these characteristics? Please add to the table.

 Response: Table 1 shows the general characteristics of the research subjects using descriptive statistics. In addition, variables such as the age, height, and weight of the research subjects were judged not to be variables that affect the dependent variable of this study, so homogeneity tests were not conducted separately. I hope for your understanding.

Clarify that the duration you mean after intervention is four weeks, to remind the reader and remain on track.

 Response: I have made corrections as per your request.

“This study randomly assigned patients undergoing total knee arthroplasty to an experimental group and a control group and applied a 4-week intervention.”

Discussion:

Virtual reality home based program. It is only mentioned in the title and the discussion, but never in the methods, so please clarify that in the methods.

Response: As requested, the term home-based has been removed to clarify the meaning.